# Organizational barriers in HPV vaccination uptake: A cross-sectional study among health sciences students

Giuseppina Palena[1]☉, Irene Stilo[1]☉, Michele Sorrentino🆔[1,2]*, Claudio Fiorilla🆔[1], Raffaele Palladino🆔[1,3,4]

**1** Department of Public Health, University "Federico II" of Naples, Naples, Italy, **2** PhD National Programme in One Health approaches to infectious diseases and life science research, Department of Public Health, Experimental and Forensic Medicine, University of Pavia, Pavia, Italy, **3** Interdepartmental Research Center in Healthcare Management and Innovation in Healthcare (CIRMIS), Naples, Italy, **4** Department of Primary Care and Public Health, School of Public Health, Imperial College, London, the United Kingdom

☉ These authors contributed equally to this work.
* michele.sorrentino@unina.it

## Abstract

### Background

Human papillomavirus (HPV) is a leading cause of cervical, anal, and oropharyngeal cancers. Despite the proven effectiveness of HPV vaccination, uptake remains low, particularly among males and specific demographic groups. In Italy, national HPV vaccination coverage stagnated at 38.8% in 2022, with significant regional disparities. While previous studies have explored individual and cultural barriers, organizational challenges within academic institutions remain underexamined. The aim of this cross-sectional study was assessing the uptake of HPV vaccination among Health Sciences students at the University of Naples Federico II.

### Methods

An anonymous online survey collected data on demographics, vaccination status, and institutional barriers. Logistic regression models were used to identify predictors of vaccination, adjusting for demographic factors, institutional communication, and knowledge-attitude scores.

### Results

Among 354 participants, 55.1% reported receiving at least one HPV vaccine dose. Female students had significantly higher vaccination rates (aOR: 7.95; 95% CI: 4.24–14.90), while older age was associated with lower uptake (aOR: 0.81; 95% CI: 0.73–0.91). Institutional vaccination invitations increased the likelihood of vaccination nearly threefold (aOR: 2.81; 95% CI: 1.48–5.33). Attitudes toward vaccination

**Data availability statement:** The raw data supporting the conclusions of this article will be made available by the authors, without undue reservation.

**Funding:** The author(s) received no specific funding for this work.

**Competing interests:** The authors have declared that no competing interests exist.

strongly predicted uptake, whereas knowledge scores showed no significant association.

## Conclusions

These findings highlight the need for structured institutional interventions, including targeted education, proactive vaccination invitations, and improved access to on-campus vaccination services. Strengthening university-led initiatives could significantly enhance HPV vaccine uptake among future healthcare professionals, contributing to broader public health efforts in HPV-related cancer prevention.

---

## 1. Introduction

Human Papillomavirus (HPV) is the most common sexually transmitted infection globally, with persistent infections causing cervical, anal, oropharyngeal, and other genital cancers [1,2]. Cervical cancer alone accounts for over 300,000 annual deaths worldwide, disproportionately affecting low- and middle-income countries [3]. HPV-related cancers represent 5% of new global cancer cases [4], with Europe reporting an estimated 58,169 new cases annually, including 17,486 cervical cancer deaths [5]. In Italy, approximately 1,500 cervical cancer deaths occur each year [6], and the economic burden of HPV-related diseases is substantial: the Italian Health System incurred direct costs of €542.7 million, a figure projected to rise by €16.2–€37.5 million with the adoption of innovative cancer therapies [6].

HPV vaccination is a proven primary prevention strategy, endorsed by the World Health Organization (WHO) since 2006 as a critical public health measure [7]. Robust evidence supports its efficacy, with studies showing vaccination reduces infections by 71–81% in vaccinated individuals and cervical cancer incidence by up to 86% in populations with high coverage [8,9]. Despite these successes, significant gaps persist. For instance, oral HPV prevalence is markedly higher in men than in women, underscoring gender-specific transmission dynamics [10]. However, even current vaccine formulations have limitations. While the nonavalent vaccine (targeting HPV6/11/16/18/31/33/45/52/58) covers 61.7% of genital infections, 38.3% are caused by non-vaccine genotypes such as HPV51, HPV53, HPV66, and HPV68. These genotypes are prevalent in both women and men, highlighting gaps in protection [10]. Notably, the nonavalent vaccine still outperforms the quadrivalent formula, particularly in males with HPV-positive partners, demonstrating broader efficacy in reducing transmission [11]. Despite these advances, Italy's vaccination efforts face systemic challenges. National HPV vaccination coverage for the complete cycle stagnated at 38.8% in 2022, with marked regional disparities and lower uptake among males [12,13].

Barriers to HPV vaccination are multifactorial, spanning individual, cultural, and systemic challenges. While prior studies emphasize parental hesitancy (e.g., safety concerns, lack of provider recommendations) and structural inequities (e.g., cost, clinic access) [14], organizational barriers—such as institutional policies, resource

allocation, and healthcare delivery inefficiencies—remain understudied. For instance, fragmented vaccine promotion, limited access in academic settings, and poor coordination between public health systems and universities may uniquely hinder uptake among young adults [15].

This study builds on prior research focused on medical students [16], expanding the scope to health sciences students a population critical to public health delivery yet underrepresented in vaccination behavior research and assessing whether similar challenges persist across different healthcare professions. Accordingly, health sciences students play a crucial role in addressing these gaps, as their vaccination behaviors not only affect their own health but also shape public trust and influence future clinical recommendations [17]. Compounding this, health sciences students often face paradoxical barriers: despite medical training, they report gaps in HPV knowledge and vaccine prioritization, suggesting institutional failures in education and resource provision [17].

This cross-sectional study examines organizational barriers to HPV vaccination uptake among health sciences students in Italy. By analyzing institutional policies, access inequities, and campus-level logistical challenges, we aim to: 1) Identify organizational barriers in vaccine delivery within a selected population; 2) Assess the impact of other factors 3) Propose interventions to strengthen campus-based vaccination infrastructure. Our findings will inform strategies to improve coverage among future healthcare professionals, leveraging their role as public health advocates to amplify population-wide prevention efforts.

## 2. Methods

### 2.1. Study design

This cross-sectional study was conducted from October to December 2024 at the University of Naples Federico II, a public university in Southern Italy. Data were collected via a self-administered, anonymous online survey distributed to all students enrolled in nursing and healthcare professions programs (N = 2745). Of these, 354 participants completed the survey.

Eligibility criteria included: enrolment in nursing or healthcare professions degree program, aged 18 or above. Participants were recruited through institutional emails, campus posters, and information booths. To minimize selection bias, the survey link was disseminated university-wide, with three reminder emails sent at weekly intervals. The target sample size was calculated using the formula for finite populations:

$$n = N \cdot Z2 \cdot p(1-p)e2(N-1) + Z2 \cdot p(1-p)n = e2(N-1) + Z2 \cdot p(1-p)N \cdot Z2 \cdot p(1-p)$$

*A 50% expected proportion (*p*\*=0.5) was assumed to ensure a conservative estimate, as no prior data on HPV vaccination uptake in this population were available. A 3% margin of error (*e*\*=0.03) and 95% confidence level ($Z$=1.96) were selected to balance precision and feasibility. Applying the formula, the minimum required sample size was 338. To account for non-response and incomplete data, we aimed for 406 participants (20% oversampling).

The study received ethical approval from the University Hospital Ethics Committee (Protocol #0014525; March 24, 2023). The study protocol was approved by the University Hospital Ethics Committee (Prot. no. 0014525, 24th of March 2023). The study was conducted in accordance with the local legislation and institutional requirements. The participants provided their written informed consent to participate in this study, obtained electronically, with consent forms stored separately from survey data to ensure anonymity.

### 2.2. Study variables

The survey was adapted from a validated HPV knowledge/attitude questionnaire [18].It underwent a content validation phase: where it was reviewed by a panel of ten physicians and public health experts; and, a pilot testing, where it was administered to 30 students to assess clarity and reliability (Cronbach's α = 0.82 for attitude scale). The survey is available in supplemental materials.

Socio-demographic characteristics included age, gender (male or female), year of study, degree program, and smoking habits (smoker, non-smoker, or former smoker). HPV vaccination status was the primary outcome (yes or no). Barriers were divided in three domains: organizational (receipt of institutional vaccination invite; yes or no), attitude (17-item scale, coded as, 1 = strongly disagree to 5 = strongly agree), and knowledge (23-item true/false assessment, coded as 1 = correct, 0 = un-correct answer).

### 2.3. Statistical Analysis

Descriptive statistics were computed to summarize participant characteristics. Continuous variables (e.g., age, knowledge scores) were expressed as mean ± standard deviation (SD) or median with interquartile range (IQR), depending on their distribution. Normality was assessed using the Shapiro-Wilk test. Categorical variables (e.g., vaccination status, sex) were reported as frequencies and percentages. Multivariable logistic regression models were constructed to identify predictors of HPV vaccination uptake. The primary model adjusted for sociodemographic and organizational covariates: sex (male/female), age (continuous), year of enrolment (1st, 2nd, 3rd, or other), smoking status (non-smoker/former smoker/current smoker), province of residence (categorical: NA1–3, AV e BV, SA, CE, other), and receipt of institutional vaccination invitations (yes/no). A sensitivity analysis expanded this model by incorporating two additional covariates: knowledge [operationalized as a continuous score (0–100%), derived from a 23-item true/false assessment. Participants with >2 missing responses were excluded from this sub analysis] and attitude [quantified as a composite score (range: 17–85), calculated by summing responses to the 17-item Likert scale (1 = strongly disagree to 5 = strongly agree)]. Results are presented as adjusted odds ratios (aORs) with 95% confidence intervals (CIs). Statistical significance was defined as $p < 0.05$. Multicollinearity between predictors was assessed using variance inflation factors (VIF), with all values <5 indicating negligible collinearity. All analyses were performed using Stata MP 18.0 (StataCorp LLC, College Station, TX).

## 3. Results

### 3.1. Demographic characteristics

The study included 354 participants, with a mean age of 22.7 years (±0.11). Females constituted 73.5% of the sample, and 55.1% reported receiving at least one dose of the HPV vaccine. Most participants were enrolled in healthcare professions (79.7%), including a subset of 72 nursing students (20.3%). Smoking habits varied: 68.4% identified as non-smokers, 25.7% as current smokers, and 5.9% as former smokers. Geographically, 70.6% resided in the provinces of Naples (NA1–3), while smaller proportions lived in Avellino/Benevento (2.5%), Salerno (5.7%), Caserta (10.5%), or other regions (10.7%). First-year students comprised 61.3% of the sample, followed by third-year (24.8%), second-year (12.8%), and sixth-year (0.3%) students (Table 1).

### 3.2. Main findings

In the primary multivariable logistic regression model (Fig 1), adjusting for sex, age, academic year, smoking status, and vaccination invitations, female sex showed the greatest likelihood of vaccination uptake (odds ratio [OR]: 7.95; 95% confidence interval [CI]: 4.24–14.90). Each additional year of age was associated with a 19% lower likelihood of vaccination (OR: 0.81; 95% CI: 0.73–0.91). Third-year students had a greater likelihood of being vaccinated compared to other cohorts (OR: 2.03; 95% CI: 1.02–4.07), and participants who received an institutional vaccination invitation were nearly three times more likely to be vaccinated (OR: 2.81; 95% CI: 1.48–5.33).

When adjusting for knowledge (23-item score) and attitudes (17-item Likert scale score) (Fig 2), the association between third-year enrollment and vaccination attenuated and was no longer statistically significant (OR: 1.99; 95% CI: 0.94–4.24). A positive attitude toward vaccination was strongly associated with a greater likelihood of uptake (OR: 2.79; 95% CI: 1.57–4.98), whereas knowledge scores showed no significant association (OR: 0.98; 95% CI: 0.95–1.01).

**Table 1. Demographic characteristics.**

| Study population | N | Percentage |
|---|---|---|
| **Sample size** | 354 | |
| **Sex** | | |
| Male | 96 | 26,65% |
| Female | 260 | 73,45% |
| **Smoking habit** | | |
| No | 232 | 68,44% |
| Former smoker | 20 | 5,90% |
| Yes | 87 | 25,66% |
| **Carrier** | | |
| Health care professions | 282 | 79,66% |
| Nursing sciences | 72 | 20,34% |
| **Year of enrollment** | | |
| 1st | 215 | 61,25% |
| 2nd | 45 | 12,82% |
| 3rd | 87 | 24,79% |
| 6th | 1 | 0,28% |
| O.P.Y. | 3 | 0.85% |
| **Vaccination Status** | | |
| No | 158 | 44,63% |
| Yes | 194 | 54,80% |
| Missing | 2 | 0,56% |
| **Province of residence** | | |
| NA 1–3 | 250 | 70,62% |
| AV e BV | 9 | 2,54% |
| SA | 20 | 5,65% |
| CE | 37 | 10,45% |
| Other | 38 | 10,73% |
| **Mean Age** | 22,65±0.11 | |

Inequalities in vaccination uptake between sexes persisted across models: females had significantly higher knowledge scores (OR: 6.8; 95% CI: 3.57–13.01) and more favorable attitudes (OR for males: 0.38; 95% CI: 0.29–0.50). Geographic origin and smoking status showed no association with vaccination outcomes in any model. Also, inverse relationship between age and vaccination likelihood remained robust (OR: 0.80; 95% CI: 0.71–0.91). The magnitude of the sex inequality slightly decreased but remained significant (OR: 6.82; 95% CI: 3.57–13.01). The effect of institutional invitations remained consistent (OR: 2.57; 95% CI: 1.33–4.97), reinforcing their role as a modifiable organizational factor. Notably, former smokers showed a non-significant tendency toward more positive vaccination attitudes (OR: 0.55; 95% CI: 0.17–1.74).

A comprehensive summary table of all statistical analyses, including significant and non-significant outcomes for both models, is provided in the Supplementary Material.

## 4. Discussion

This cross-sectional study identifies key predictors of HPV vaccination uptake among 354 health sciences students in Italy, revealing that female sex, institutional vaccination invitations, and positive attitudes—rather than

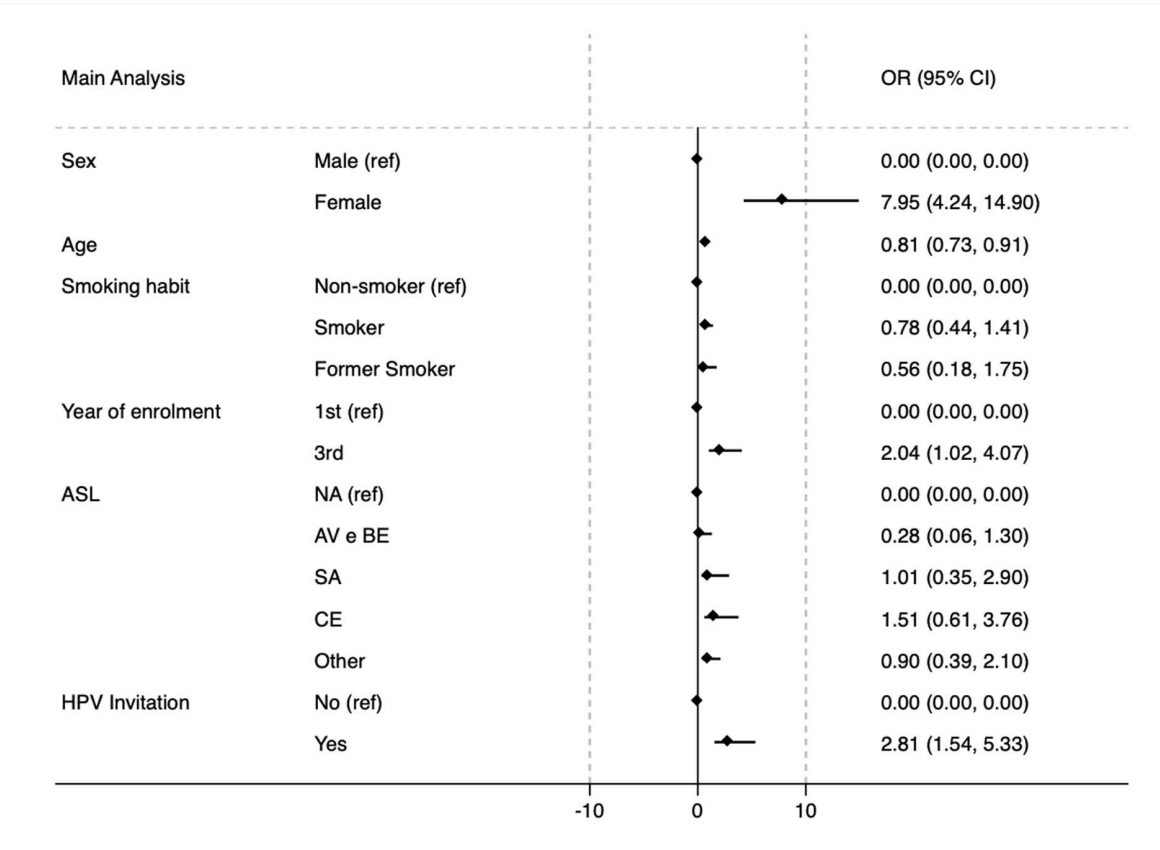

**Fig 1. Main analysis.** Primary multivariable logistic regression model, adjusted for sex, age, academic year, smoking status, and vaccination invitations.

knowledge—drive adherence, while older age reduces likelihood. These findings join our previous research on medical students [12], where clinical knowledge and academic progression were central. By broadening the focus to health sciences students —an understudied cohort critical to public health—we uncover profession-specific barriers: health sciences students rely more on institutional outreach and personal attitudes, with knowledge playing no significant role. Such disparities challenge generic educational campaigns and underscore the need for tailored, multidisciplinary strategies to improve HPV coverage across healthcare teams, addressing systemic inequities in academic and clinical settings. Female students exhibited an eightfold higher likelihood of HPV vaccination compared to males, aligning with global trends where women are more proactive in preventive health measures [19]. This disparity likely stems from historical policies: Italy's HPV vaccination program initially targeted females, integrating the vaccine into cervical cancer screening campaigns with structured catch-up strategies [20]. Consequently, males may perceive HPV as a "female-specific" issue, compounded by limited awareness of its association with non-cervical cancers [21]. Addressing this requires gender-neutral public health messaging and equitable access to vaccination for all students. Each additional year of age reduced vaccination likelihood by 19%, consistent with studies showing younger individuals are more receptive to health innovations due to cognitive adaptability [22] and targeted educational exposure [23]. Third-year students demonstrated higher uptake, but this association attenuated after adjusting for knowledge/attitudes, suggesting academic progression indirectly enhances awareness. This underscores the need for early, curriculum-integrated HPV education to capitalize on younger students' adaptability.

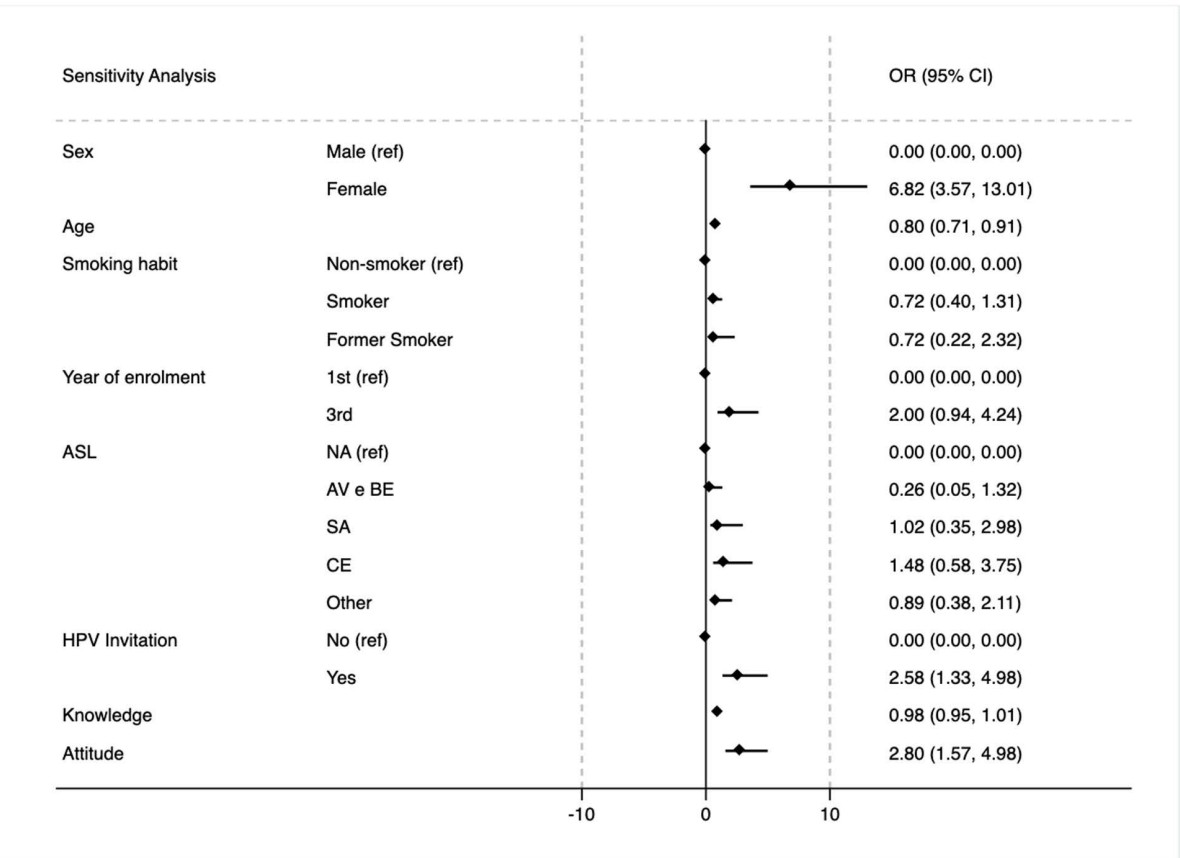

**Fig 2. Sensitivity analysis.** Secondary multivariable logistic regression model, adjusted for for knowledge (23-item score) and attitudes (17-item Likert scale score).

Receiving a vaccination invitation nearly tripled the odds of uptake, reinforcing the efficacy of structured organizational interventions. Proactive outreach—such as automated reminders, peer-led campaigns, and on-campus vaccination clinics—reduces logistical barriers and normalizes vaccine uptake [24,25]. Universities, as hubs of health advocacy, should institutionalize these strategies through partnerships with local health authorities. Positive attitudes toward vaccination were strongly associated with uptake, whereas knowledge scores showed no significant link. This aligns with behavioral theories (e.g., Health Belief Model) where perceived benefits and cues to action outweigh factual knowledge [26]. Interventions must therefore pair education with trust-building initiatives, such as testimonials from vaccinated peers or faculty.

Furthermore, our data highlight a significant evolution in parental attitudes towards HPV vaccination over time. This is evidenced by higher vaccination rates among younger cohorts compared to older age groups, reflecting increased parental acceptance and proactive decision-making in recent years [27]. However, several barriers persist, including limited awareness, logistical constraints, and reduced accessibility for students residing outside the region.

Former smokers showed a non-significant trend toward favorable vaccination attitudes, contrasting with studies linking smoking to vaccine hesitancy [28,29]. While underpowered in this cohort, this finding warrants exploration of whether health-conscious behavior changes (e.g., smoking cessation) correlate with greater vaccine acceptance. Among school nurses, a positive stance on HPV vaccination is largely influenced by their perceived role as opinion leaders in promoting the vaccine [30]. Additionally, parental influence remains a key factor in vaccine uptake among young adult college

students [31]. These findings highlight the need for targeted interventions that not only educate but also foster positive attitudes and leverage key influencers to enhance HPV vaccination rates.

Awareness of HPV and its link to multiple cancers, including cervical and throat cancers, remains low, even among vaccinated individuals [32]. Healthcare professionals are pivotal in addressing concerns and advocating for vaccination [33]. Therefore, comprehensive education and targeted interventions are crucial to improving HPV vaccination rates.

Collectively, these findings underscore the interplay of demographic predispositions, institutional strategies, and psychosocial factors in shaping HPV vaccine uptake. While individual traits like sex and age are immutable, organizational interventions—such as targeted invitations—represent modifiable levers for improving adherence.

## 5. Policy

Health science students represent a key population for examining the factors influencing HPV vaccination uptake, as their advanced education and medical knowledge should ideally correlate with higher vaccination rates [34]. The decline in vaccination rates, further exacerbated by the COVID-19 pandemic [35], poses a significant public health risk, potentially leading to a rise in HPV-related cancers and increased healthcare costs. Expanding free-of-charge HPV vaccination programs to include males has been recognized as a cost-effective strategy for reducing the burden of cervical cancer [36], particularly given findings that male partners of HPV-positive women often harbor high-risk genotypes linked to cervical lesions [37]. Targeted educational campaigns should address these risks, emphasizing that vaccinati ng men not only protects them but also reduces transmission to female partners, as shown in studies where 62.5% of male infections involved genotypes covered by the nonavalent vaccine [37].

To address gender disparities, targeted educational campaigns should be developed to improve awareness among male students regarding HPV-related risks and the benefits of vaccination.

In Italy, the latest national vaccination program, PNPV 2023–2025, has expanded free-of-charge HPV vaccination recall efforts, aiming to increase coverage rates [38] prioritizing the nonavalent vaccine due to its broader coverage of high-risk HPV types and potential to prevent up to 90% of high-grade cervical lesions, reducing long-term healthcare burdens [39]. While this initiative is expected to improve adherence, its full impact requires further evaluation. Additionally, updating communication strategies to leverage digital media and social platforms could be crucial in reaching unvaccinated individuals who are less responsive to traditional public health messaging [40]. Finally, continuous monitoring and assessment of these interventions will be essential to refine policies, enhance vaccine accessibility, and ensure equitable and widespread HPV vaccination uptake, ultimately improving long-term public health outcomes.

To bridge existing coverage gaps, it is essential to implement comprehensive strategies, including catch-up programmes for individuals who missed earlier doses and the integration of educational campaigns into university curricula. Expanding free vaccination access to both residents and non-residents could further enhance uptake. Additionally, leveraging modern communication channels to engage resistant subgroups and delivering tailored educational interventions to improve awareness of HPV risks and vaccination benefits will be critical.

Ultimately, a multifaceted approach that combines policy changes, improved accessibility, and targeted education is necessary to overcome vaccination barriers and reduce the long-term burden of HPV-related cancers [41].

## 6. Strengths and limitations

This study offers novel insights into organizational barriers influencing HPV vaccination uptake among health sciences students, a population critical to shaping future public health practices. Its strengths include the use of a validated questionnaire to ensure methodological rigor, multivariable logistic regression to adjust for confounders such as age and institutional invitations, and a sample size sufficient to detect meaningful associations. By focusing on understudied systemic factors—such as institutional outreach and geographic accessibility—this work advances understanding of how organizational policies, rather than individual hesitancy alone, shape vaccine adherence. The inclusion of both medical

and non-medical healthcare students enhances relevance across academic disciplines, while the emphasis on gender disparities aligns with global calls for equitable vaccination strategies.

However, several limitations warrant consideration. The cross-sectional design precludes causal inferences, limiting our ability to determine whether organizational interventions (e.g., vaccination invitations) directly drive uptake or reflect broader institutional priorities. Self-reported vaccination status, though practical, risks recall or social desirability bias, particularly in a population aware of public health norms. Unmeasured variables, such as socioeconomic status or prior healthcare access, may further confound observed associations. While academic year served as a proxy for HPV-related curriculum exposure, extracurricular influences (e.g., peer discussions, clinical rotations) were not assessed, potentially overlooking informal educational pathways.

Geographic generalizability is constrained by the single-university setting, as regional policies (e.g., Italy's PNPV 2023–2025) and local healthcare infrastructure may uniquely shape accessibility. For instance, non-resident students' reduced access to regional programs, highlighted in our findings, may not reflect challenges in other jurisdictions. Additionally, the study's timing during the COVID-19 pandemic—a period of disrupted healthcare services—may conflate organizational barriers with pandemic-specific hesitancy, though this contextualizes urgent policy needs.

Future research should adopt longitudinal designs to disentangle temporal trends, incorporate multi-institutional cohorts to assess geographic variability, and evaluate emerging strategies like digital outreach campaigns. Despite these constraints, the study provides actionable evidence for universities and policymakers to refine institutional vaccine delivery systems.

## 7. Conclusion

The findings of this study highlight the importance of targeted interventions in increasing HPV vaccination uptake among health profession students. Despite the availability of the vaccine, gaps in awareness, attitudes, and structural barriers continue to hinder optimal coverage, particularly among male students. Expanding recall strategies, improving educational campaigns, and integrating behavioral science principles into vaccination programs are crucial steps toward improving adherence. Public health authorities should ensure that HPV vaccination is easily accessible and systematically promoted among all eligible populations. Moreover, ongoing monitoring of vaccination trends and the impact of policy changes—such as the extended recall program in Italy—will be essential in assessing progress toward HPV-related cancer prevention. Future research should focus on identifying additional behavioral and systemic factors that influence vaccination uptake, ensuring that all individuals, regardless of gender or background, can benefit from HPV prevention efforts.

## Supporting information

**S1 Table. Supplementary material.**
(DOCX)

**S1 File. Inclusivity-in-global-research-questionnaire.**
(DOCX)

**S2 File. Database.**
(XLSX)

## Author contributions

**Conceptualization:** Michele Sorrentino.

**Data curation:** Giuseppina Palena, Irene Stilo.

**Formal analysis:** Michele Sorrentino.

**Methodology:** Giuseppina Palena, Irene Stilo, Michele Sorrentino.

**Supervision:** Raffaele Palladino.

**Validation:** Raffaele Palladino.

**Writing – original draft:** Giuseppina Palena, Irene Stilo, Michele Sorrentino, Claudio Fiorilla, Raffaele Palladino.

**Writing – review & editing:** Michele Sorrentino, Claudio Fiorilla, Raffaele Palladino.

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
