## [Decision Letter · Decision Letter 0]

Dear Dr. Sorrentino,

Thank you for submitting your manuscript to PLOS ONE. After careful consideration, we feel that it has merit but does not fully meet PLOS ONE’s publication criteria as it currently stands. Therefore, we invite you to submit a revised version of the manuscript that addresses the points raised during the review process.

One of the three reviewers reports that your article is similar to an article already published in Frontiers in Public Health journal,

Front. Public Health, 17 November 2023; Sec. Infectious Diseases: Epidemiology and Prevention

Volume 11 - 2023 | https://doi.org/10.3389/fpubh.2023.1272630

I ask the authors to reply to the Reviewer commenting on the differences between the two articles.

In addition, I report the following points to improve your manuscript,

1) lines 77-80. Authors should provide information on how the expected proportion and margin of error values were identified.

2) To help readers, a summary table with all the statistical tests performed, both significant and non-significant, would be necessary.

We look forward to receiving your revised manuscript.

Kind regards,

Nicola Serra

Academic Editor

PLOS ONE

Reviewers' comments:

Reviewer's Responses to Questions

**Comments to the Author**

1. Is the manuscript technically sound, and do the data support the conclusions?

Reviewer #1: Yes

Reviewer #2: Yes

Reviewer #3: Yes

2. Has the statistical analysis been performed appropriately and rigorously?

Reviewer #1: Yes

Reviewer #2: I Don't Know

Reviewer #3: I Don't Know

3. Have the authors made all data underlying the findings in their manuscript fully available?

Reviewer #1: Yes

Reviewer #2: Yes

Reviewer #3: Yes

4. Is the manuscript presented in an intelligible fashion and written in standard English?

Reviewer #1: Yes

Reviewer #2: Yes

Reviewer #3: Yes

Reviewer #1: This work utilises methods, and arrives at conclusions, that are sufficiently similar to a previous study published by the same authors that this manuscript provides little to no new insight into the purported area of investigation.

Reviewer #2: This manuscript proposes an in-depth analysis of organizational barriers to HPV vaccination uptake among health sciences students in Italy. The paper is well written and discusses the importance of targeted interventions in increasing HPV vaccination uptake among health professions students.

Given the purpose of the study, it would have been appropriate to include more male students in the sample. This would have provided a better understanding of the higher likelihood of HPV vaccination among female students.

I only give a few tips to improve the manuscript.

ABSTRACT:

Pag. 7, lines 19-20. This cross-sectional study assessed HPV vaccination uptake among health sciences students at the University of Naples Federico II.

Please consider moving the aim of the study in the background section of the abstract.

INTRODUCTION:

Pag. 9, line 62. 2) Assess the impact of other factors.

Please consider listing the potential factors.

It seems that your introduction is too short. Please emphasize more topics, such as HPV prevalence and vaccines impact on the population. I invite the authors to consider the following works:

Buttà M. et al. Orogenital Human Papillomavirus Infection and Vaccines: A Survey of High- and Low-Risk Genotypes Not Included in Vaccines. Vaccines (Basel). 2023

Bosco L. et al. Potential impact of a nonavalent anti HPV vaccine in Italian men with and without clinical manifestations. Sci Rep. 2021

MATERIALS AND METHODS:

Pag. 9, lines 68-71. This cross-sectional study was conducted from October to December 2024 at the University of Naples Federico II, a public university in Southern Italy. Data were collected via a self-administered, anonymous online survey distributed to all students enrolled in nursing and healthcare professions programs (N= 2745).

Please consider writing also the number of the participants effectively recruited.

Reviewer #3: Despite the availability of effective HPV vaccines and an appropriate Italian vaccination programme, vaccination coverage remains extremely low. It is therefore imperative to analyse the reasons for the low uptake of the vaccine and to identify ways to improve coverage.

In this scenario, the authors conducted a clear and well-designed analysis of the factors (e.g. organisational, demographic and attitudinal) affecting HPV vaccination coverage among 354 health science students in Italy. The paper is well-written, and the analysis performed leads to clear and sound results.

May I suggest you the following paper focusing on analyzing the potential use of HPV vaccines?

Sucato A, Serra N, Buttà M, Gregorio LD, Pistoia D, Capra G. Human Papillomavirus Infection in Partners of Women Attending Cervical Cancer Screening: A Pilot Study on Prevalence, Distribution, and Potential Use of Vaccines. Vaccines (Basel). 2025 Feb 11;13(2):172. doi: 10.3390/vaccines13020172.

Capra G, Giovannelli L, Matranga D, Bellavia C, Guarneri MF, Fasciana T, Scaduto G, Firenze A, Vassiliadis A, Perino A. Potential impact of a nonavalent HPV vaccine on HPV related low-and high-grade cervical intraepithelial lesions: A referral hospital-based study in Sicily. Hum Vaccin Immunother. 2017 Aug 3;13(8):1839-1843. doi: 10.1080/21645515.2017.1319026.

**Do you want your identity to be public for this peer review?** For information about this choice, including consent withdrawal, please see our Privacy Policy

Reviewer #1: No

Reviewer #2: No

Reviewer #3: No

---

## [Author Response · Author response to Decision Letter 1]

26 May 2025

Dear Editors,

Please find enclosed a revised version of our manuscript. We have sought to address all Reviewers comments and believe that the paper is greatly improved. Please find our responses to individual comments below.

Dear Authors,

One of the three reviewers reports that your article is similar to an article already published in Frontiers in Public Health journal,

Front. Public Health, 17 November 2023; Sec. Infectious Diseases: Epidemiology and Prevention

Volume 11 - 2023 | https://doi.org/10.3389/fpubh.2023.1272630

I ask the authors to reply to the Reviewer commenting on the differences between the two articles.

In addition, I report the following points to improve your manuscript,

lines 77-80. Authors should provide information on how the expected proportion and margin of error values were identified.

Thank you for raising this important point. Below, we clarify how the expected proportion (*p*) and margin of error (*e*) were determined for the sample size calculation:

When prior data on HPV vaccination uptake in the target population (Health Sciences students in Southern Italy) are unavailable, a proportion of 50% (*p* = 0.5) is conventionally used in sample size calculations. This value maximizes the variance (*p*(1−*p*)), yielding the most conservative (largest) sample size estimate, ensuring robustness across potential true proportions.

A 3% margin of error was selected to balance precision and feasibility. This is standard in epidemiological studies aiming for high confidence in prevalence estimates while maintaining practical recruitment goals.

A 95% confidence level aligns with common practice in health sciences research, reflecting a 5% acceptable Type I error rate.

We have updated the Materials and Methods section to explicitly clarify the rationale behind the expected proportion, margin of error, and confidence level used in the sample size calculation. The revised paragraph now reads:

"A 50% expected proportion (p* = 0.5) was assumed to ensure a conservative estimate, as no prior data on HPV vaccination uptake in this population were available. A 3% margin of error (*e* = 0.03) and 95% confidence level (Z = 1.96) were selected to balance precision and feasibility. Applying the formula, the minimum required sample size was 338."*

This addition enhances transparency and aligns with your suggestion to justify the parameters used in the calculation.

We appreciate your thorough review and are happy to address any further questions or adjustments.

To help readers, a summary table with all the statistical tests performed, both significant and non-significant, would be necessary.

We thank the editor for this suggestion. As requested, a summary table detailing all statistical tests performed in this study, including significant and non-significant associations across both models, has been included as Supplementary Material. To ensure transparency and facilitate reader interpretation, we have added the following clarifying statement to the results section:

“A comprehensive summary table of all statistical analyses, including significant and non-significant outcomes for both models, is provided in the Supplementary Material.”

Reviewer #1: This work utilises methods, and arrives at conclusions, that are sufficiently similar to a previous study published by the same authors that this manuscript provides little to no new insight into the purported area of investigation.

Thank you for your thoughtful critique. We have revised the Introduction to explicitly clarify the unique contribution of this study compared to our prior work. Specifically, we added in the Introduction the sentence below:

"This study builds on prior research focused on medical students [12] expanding the scope to health sciences students—including nursing and allied healthcare trainees—a population critical to public health delivery yet underrepresented in vaccination behavior research and assessing whether similar challenges persist across different healthcare professions."

Also, in the Discussion, we refined the concluding paragraph to emphasize key contrasts:

“This cross-sectional study identifies key predictors of HPV vaccination uptake among 354 health sciences students in Italy, revealing that female sex, institutional vaccination invitations, and positive attitudes—rather than knowledge—drive adherence, while older age reduces likelihood. These findings join our previous research on medical students [12], where clinical knowledge and academic progression were central. By broadening the focus to nursing and allied health trainees—an understudied cohort critical to public health—we uncover profession-specific barriers: health sciences students rely more on institutional outreach and personal attitudes, with knowledge playing no significant role. Such disparities challenge generic educational campaigns and underscore the need for tailored, multidisciplinary strategies to improve HPV coverage across healthcare teams, addressing systemic inequities in academic and clinical settings.”

These revisions explicitly position the current study as a necessary extension of prior research, highlighting novel mechanisms (e.g., attitudes over knowledge) and population-specific insights (health sciences vs. medical students). We hope this clarifies the originality and public health relevance of our work.

Thank you again for your rigorous feedback. Please let us know if further adjustments would strengthen the manuscript.

Reviewer #2: This manuscript proposes an in-depth analysis of organizational barriers to HPV vaccination uptake among health sciences students in Italy. The paper is well written and discusses the importance of targeted interventions in increasing HPV vaccination uptake among health professions students.

Given the purpose of the study, it would have been appropriate to include more male students in the sample. This would have provided a better understanding of the higher likelihood of HPV vaccination among female students.

I only give a few tips to improve the manuscript.

ABSTRACT:

Pag. 7, lines 19-20. This cross-sectional study assessed HPV vaccination uptake among health sciences students at the University of Naples Federico II.

Please consider moving the aim of the study in the background section of the abstract.

Thank you for your valuable feedback. We have moved the study aim to the Background section of the abstract and revised the phrasing for improved clarity and grammatical correctness. The updated sentence now reads:

"The aim of this cross-sectional study was to assess the uptake of HPV vaccination among Health Sciences students at the University of Naples Federico II."

This adjustment ensures better readability and aligns with standard academic structure. We appreciate your thoughtful suggestion.

INTRODUCTION:

Pag. 9, line 62. 2) Assess the impact of other factors. Please consider listing the potential factors.

It seems that your introduction is too short. Please emphasize more topics, such as HPV prevalence and vaccines impact on the population. I invite the authors to consider the following works:

Buttà M. et al. Orogenital Human Papillomavirus Infection and Vaccines: A Survey of High- and Low-Risk Genotypes Not Included in Vaccines. Vaccines (Basel). 2023

Bosco L. et al. Potential impact of a nonavalent anti HPV vaccine in Italian men with and without clinical manifestations. Sci Rep. 2021

We sincerely thank the reviewer for their insightful feedback and for highlighting the importance of expanding the discussion on HPV prevalence and vaccine impact. In response to your suggestions, we have integrated additional data into the introduction to better emphasize these topics, as follows:

“Cervical cancer alone accounts for over 300,000 annual deaths worldwide, disproportionately affecting low- and middle-income countries [3]. HPV-related cancers represent 5% of new global cancer cases [4], with Europe reporting an estimated 58,169 new cases annually, including 17,486 cervical cancer deaths [5]. In Italy, approximately 1,500 cervical cancer deaths occur each year [6], and the economic burden of HPV-related diseases is substantial: the Italian Health System incurred direct costs of €542.7 million, a figure projected to rise by €16.2–€37.5 million with the adoption of innovative cancer therapies [6].

HPV vaccination is a proven primary prevention strategy, endorsed by the World Health Organization (WHO) since 2006 as a critical public health measure [7]. Robust evidence supports its efficacy, with studies showing vaccination reduces infections by 71–81% in vaccinated individuals and cervical cancer incidence by up to 86% in populations with high coverage [8], [9]. Despite these successes, significant gaps persist. For instance, oral HPV prevalence is markedly higher in men than in women, underscoring gender-specific transmission dynamics [10]. However, even current vaccine formulations have limitations. While the nonavalent vaccine (targeting HPV6/11/16/18/31/33/45/52/58) covers 61.7% of genital infections, 38.3% are caused by non-vaccine genotypes such as HPV51, HPV53, HPV66, and HPV68. These genotypes are prevalent in both women and men, highlighting gaps in protection [10]. Notably, the nonavalent vaccine still outperforms the quadrivalent formula, particularly in males with HPV-positive partners, demonstrating broader efficacy in reducing transmission [11]. Despite these advances, Italy’s vaccination efforts face systemic challenges. National HPV vaccination coverage for the complete cycle stagnated at 38.8% in 2022, with marked regional disparities and lower uptake among males [12], [13].”

We deeply appreciate your recommendation of these key references, which have enriched our analysis of gaps in current vaccine coverage and regional challenges in Italy.

MATERIALS AND METHODS:

Pag. 9, lines 68-71. This cross-sectional study was conducted from October to December 2024 at the University of Naples Federico II, a public university in Southern Italy. Data were collected via a self-administered, anonymous online survey distributed to all students enrolled in nursing and healthcare professions programs (N= 2745).

Please consider writing also the number of the participants effectively recruited.

Thank you for your insightful suggestion. The total number of participants effectively recruited (N = 354) is explicitly stated in the Results section under the Demographic characteristics subsection. To ensure clarity and avoid redundancy, we have maintained this placement, as the Results section is the standard location for reporting final sample sizes in cross-sectional studies. However, to give more focus in the Materials and Methods section, we have included a brief mention there as well (e.g. ‘Of these, 354 completed the survey’).

Reviewer #3: Despite the availability of effective HPV vaccines and an appropriate Italian vaccination programme, vaccination coverage remains extremely low. It is therefore imperative to analyse the reasons for the low uptake of the vaccine and to identify ways to improve coverage.

In this scenario, the authors conducted a clear and well-designed analysis of the factors (e.g. organisational, demographic and attitudinal) affecting HPV vaccination coverage among 354 health science students in Italy. The paper is well-written, and the analysis performed leads to clear and sound results.

May I suggest you the following paper focusing on analyzing the potential use of HPV vaccines?

Sucato A, Serra N, Buttà M, Gregorio LD, Pistoia D, Capra G. Human Papillomavirus Infection in Partners of Women Attending Cervical Cancer Screening: A Pilot Study on Prevalence, Distribution, and Potential Use of Vaccines. Vaccines (Basel). 2025 Feb 11;13(2):172. doi: 10.3390/vaccines13020172.

Capra G, Giovannelli L, Matranga D, Bellavia C, Guarneri MF, Fasciana T, Scaduto G, Firenze A, Vassiliadis A, Perino A. Potential impact of a nonavalent HPV vaccine on HPV related low-and high-grade cervical intraepithelial lesions: A referral hospital-based study in Sicily. Hum Vaccin Immunother. 2017 Aug 3;13(8):1839-1843. doi: 10.1080/21645515.2017.1319026.

We sincerely thank the reviewer for their insightful comments and for suggesting the inclusion of the two pivotal studies by Sucato et al. (2025) and Capra et al. (2017). As requested, we have integrated these references into the revised manuscript, specifically in the "Policy " section. These studies strengthen our discussion on cost-effective strategies to improve vaccination coverage, particularly by addressing gender disparities and prioritizing nonavalent vaccines in national programs like Italy’s PNPV 2023-2025. The revisions underscore the importance of male inclusion and updated vaccine formulations in closing coverage gaps and reducing long-term HPV-related cancer burdens.

Thank you again for your valuable feedback, which has significantly enhanced the depth and relevance of our analysis.

---

## [Decision Letter · Decision Letter 1]

Organizational Barriers in HPV vaccination uptake: a cross-sectional study among health sciences students

PONE-D-25-16271R1

Dear Dr. Sorrentino,

We’re pleased to inform you that your manuscript has been judged scientifically suitable for publication and will be formally accepted for publication once it meets all outstanding technical requirements.

Kind regards,

Nicola Serra

Academic Editor

PLOS ONE

Additional Editor Comments (optional):

Reviewers' comments:

Reviewer's Responses to Questions

**Comments to the Author**

Reviewer #2: All comments have been addressed

Reviewer #3: All comments have been addressed

2. Is the manuscript technically sound, and do the data support the conclusions?

Reviewer #2: Yes

Reviewer #3: Yes

3. Has the statistical analysis been performed appropriately and rigorously?

Reviewer #2: I Don't Know

Reviewer #3: I Don't Know

4. Have the authors made all data underlying the findings in their manuscript fully available?

Reviewer #2: Yes

Reviewer #3: Yes

5. Is the manuscript presented in an intelligible fashion and written in standard English?

Reviewer #2: Yes

Reviewer #3: Yes

Reviewer #2: The manuscript has been significantly improved and is now ready for publication. I don't have any suggestions for the authors.

Reviewer #3: (No Response)

**Do you want your identity to be public for this peer review?** For information about this choice, including consent withdrawal, please see our Privacy Policy

Reviewer #2: No

Reviewer #3: No

---

## [Editor Report · Acceptance letter]

PONE-D-25-16271R1

PLOS ONE

Dear Dr. Sorrentino,

I'm pleased to inform you that your manuscript has been deemed suitable for publication in PLOS ONE. Congratulations! Your manuscript is now being handed over to our production team.

Kind regards,

on behalf of

Dr. Nicola Serra

Academic Editor

PLOS ONE